# Homocysteine in Neurology: A Possible Contributing Factor to Small Vessel Disease

**DOI:** 10.3390/ijms22042051

**Published:** 2021-02-19

**Authors:** Rita Moretti, Mauro Giuffré, Paola Caruso, Silvia Gazzin, Claudio Tiribelli

**Affiliations:** 1Department of Medical, Surgical and Health Sciences, University of Trieste, 34149 Trieste, Italy; gff.mauro@gmail.com (M.G.); paolacaruso83@gmail.com (P.C.); 2Italian Liver Foundation, AREA SCIENCE PARK, 34149 Trieste, Italy; silvia.gazzin@fegato.it (S.G.); ctliver@fegato.it (C.T.)

**Keywords:** homocysteine, SVD, neurodegeneration, neuroinflammation, oxidative stress

## Abstract

Homocysteine (Hcy) is a sulfur-containing amino acid generated during methionine metabolism, accumulation of which may be caused by genetic defects or the deficit of vitamin B12 and folate. A serum level greater than 15 micro-mols/L is defined as hyperhomocysteinemia (HHcy). Hcy has many roles, the most important being the active participation in the transmethylation reactions, fundamental for the brain. Many studies focused on the role of homocysteine accumulation in vascular or degenerative neurological diseases, but the results are still undefined. More is known in cardiovascular disease. HHcy is a determinant for the development and progression of inflammation, atherosclerotic plaque formation, endothelium, arteriolar damage, smooth muscle cell proliferation, and altered-oxidative stress response. Conversely, few studies focused on the relationship between HHcy and small vessel disease (SVD), despite the evidence that mice with HHcy showed a significant end-feet disruption of astrocytes with a diffuse SVD. A severe reduction of vascular aquaporin-4-water channels, lower levels of high-functioning potassium channels, and higher metalloproteinases are also observed. HHcy modulates the N-homocysteinylation process, promoting a pro-coagulative state and damage of the cellular protein integrity. This altered process could be directly involved in the altered endothelium activation, typical of SVD and protein quality, inhibiting the ubiquitin-proteasome system control. HHcy also promotes a constant enhancement of microglia activation, inducing the sustained pro-inflammatory status observed in SVD. This review article addresses the possible role of HHcy in small-vessel disease and understands its pathogenic impact.

## 1. Introduction

This article aims to define the role of Hcy in the development of small vessel disease (SVD) and neurological damage. We searched MEDLINE using the search terms “vascular dementia,” “subcortical vascular dementia,” “vascular cognitive impairment,” “small vessel disease,” “arteriolosclerosis,” “cerebral flow regulation,” “homocysteine” “neurovascular coupling,” “endothelium,” smooth muscle cells arteries”, “neuroinflammation” “oxidative damage”, and “neurodegeneration”. Publications were selected mostly from the past 25 years (1 January 1995 to 15 December 2020) but did not exclude frequently referenced and highly regarded older publications. The research has been extended with the same strings to EMBASE, COCHRANE LIBRARY, and LILACS databases. We have considered papers published in English, French, German, and Italian. Secondary searching was performed using the most relevant articles (following PRISMA statement, 2009) [1]. Congress abstracts and isolated case reports were not considered. We (all the Authors who contributed to the research strategy) searched the reference lists of articles identified by this search strategy and selected those we judged relevant. Review articles and book chapters are cited for providing additional details. A total of 1234 studies showed up, and appropriate studies (*n* = 312) were included. The authors carefully read all the eligible articles (Figure 1).

## 2. Cerebral Small Vessel Disease

Cerebral small vessel disease (SVD) primarily distresses the small perforating arteries, defined as vessels with less than 50 μm diameters, which perfuses the deep brain structures, the meningeal space, and the white substance [2,3]. SVD progression leads to the condition known as subcortical vascular dementia (sVAD), one of the most common forms of degenerative disorders globally, accounting for 45% of dementia cases in the world [4,5,6]. While affecting the small arteries, SVD contributes to enlargement and a loss of function of perivascular spaces (PVS) [7,8], critical in catabolic/glynphatic responses [9,10,11,12,13], as well as occlusion of small draining veins [14], with a disruption of the blood–brain barrier (BBB) [4]. The sum of all the events promotes a chronic inflammatory status, which is the pathological basis of SVD [12,15,16,17]. Small arteries undergo a pathological process named arteriolosclerosis [4,9,18,19,20], which primarily impedes the autoregulation of cerebral blood flow (CBF) exerted by small arteries [9,21,22,23,24]. Arteriolosclerosis occurs in two primary histological forms: hyperplastic and hyaline [25,26,27,28]. A reduction of the arterial elasticity, a loss of control of the resting flow, and decreased perfusion pressure towards the profound arteries are observed [29,30,31,32,33,34,35].

In the animal SVD models, there is a reduction of vasopressin and histamine, a direct consequence of the progressive disruption of neural tracts, extending from the supra-optic and tuberomammillary nuclei to the basal forebrain [9,36,37,38]. A super-imposed endothelium-mediated altered baroreflex activity is associated with low-level functions of the autonomic nervous system [39,40,41,42]. The main consequence is a decline of the cerebral blood flow control, altering the retrograde vasodilatation system [9,43,44,45,46,47,48,49,50,51,52,53,54,55,56,57,58,59,60,61,62,63]. The chronic hypoperfusion condition promotes a chronic inflammatory status, induced by glynphatic, veins, and BBB disruption. 

In SVD, there is a severe oligodendrocyte degeneration, a microglial activation (testified by a severe increase of caspase 3-RNA and matrix-metalloprotease 2 (MMP-2) expression), massive calcium inflows, and apoptosis process [34,35]. Astrocytes respond in a two-way system: in the first ischemic period, they proliferate [35,58], but when the neuroinflammation endures, they lose their end-feet, degenerate, and rapidly die [35,57,58,59,60]. Astrocytic death corresponds to an altered neurovascular coupling and a consequent induction of neuronal death [4,9,64,65,66,67,68,69,70,71]. 

The endothelium is indirectly affected by the apparent mitochondrial damage [72,73], mainly due to a loss of response to the main endothelium-derived nitric oxide-vasodilators [74], prostacyclin [75], and endothelium-derived hyperpolarizing factors (EDHF) [76,77,78,79,80,81]. The general neuroinflammation present in SVD determines a hyperproduction of peroxynitrite [82,83]. It depends on an altered redox response associated with a reduction of the activity of endothelial NO synthase (eNOS) and downregulation of the Rho-associated protein kinase (ROCK), which usually promotes the vascular endothelium growth factor (VEGF) in response to vascular injury [84,85,86,87,88,89,90,91,92,93]. 

NO is a significant mediator of vasodilatation through cGMP/PKG signals, leading to decreased Ca2+ concentration. Besides, NO-mediated signals trigger an increase in myosin light-chain phosphatase (MLCP) activity. ROCK inactivates MLCP via calcium desensitization [84,94] and therefore decreases the availability of NO [95,96]. 

Altered endothelium activation is not specific to gray matter [71,97,98,99], but is more pronounced in white matter, putamen, caudate, and in all the basal forebrain-frontal subcortical networks [100,101,102,103,104,105,106,107,108,109]. 

Taken together, SVD is a progressive disease [3,8,38,110,111,112,113,114,115,116,117,118,119], though the exact timing of its progression is not established [9]. The rapid confluence of the isolated white matter lesions, the number of silent infarcts, and the vascular lacunar events are essential in determining the cognitive and behavior impairment during SVD [4,9,38,117,118,119,120,121,122,123,124,125,126,127,128,129,130,131] (Figure 2).

## 3. Homocysteine and Brain 

Hcy is a sulfur-containing intermediary amino acid [132], recycled via the remethylation pathway or converted into cysteine via the trans-sulfuration pathway [4]. 

The methionine synthesis occurs when there is a reduction of 5,10-methylenetetrahydrofolate to 5-methyltetrahydrofolate (5-methylTHF) [133,134,135]. In remethylation, Hcy acquires a methyl group from N-5-methyltetrahydrofolate or from betaine to form methionine. The reaction with N-5-methyltetrahydrofolate occurs in all tissues and is vitamin B12-dependent. In particular, methionine adenosyltransferase (MAT) catalyzes S-adenosylmethionine (AdoMet) (SAM), actively consuming Adenosyn triphosphate (ATP) [133,134]. SAM is the methyl group donor in numerous methylation reactions, a fundamental process for the protein, phospholipid, and biogenic amines synthesis [136,137,138,139,140,141]. Every reaction made by methyltransferases produces S-adenosylhomocysteine (AdoHcy) (SAH) [142,143,144,145]. The SAM to SAH ratio defines the cell’s methylation potential [146,147,148,149,150,151,152,153,154,155]. 

In the trans-sulfuration pathway, Hcy condenses with serine to form cystathionine. It is an irreversible reaction catalyzed by the pyridoxal-50-phosphate (PLP)-containing enzyme, cystathionine β-synthase. Cystathionine is hydrolyzed by a second PLP-containing enzyme, γ-cystathionase, to form cysteine and α-ketobutyrate [144]. Excess cysteine is oxidized to taurine or inorganic sulfates or is excreted in the urine [144]. Therefore, the trans-sulfuration pathway catabolizes excess homocysteine, which is not required for methyl transfer [144,151,152,153,154,155]. 

The intrinsic capacity to differentiate between the remethylation and trans-sulfuration pathways to adapt to different intake-methionine levels in the diet strongly implies the existence of a coordinate regulation between these two pathways [144]. SAM could act as an allosteric inhibitor of methylenetetrahydrofolate reductase (MTHFR). It could also play a role as an activator of cystathionine β-synthase, promoting the trans-sulfuration pathway (cystathionine synthesis) [144]. When the methionine supply is low, there is an elevated rate of N-5-methyltetrahydrofolate production. Thus, remethylation will be favored over trans-sulfuration because the concentration of SAM is too low to activate the cystathionine β-synthase enzyme [144]. Remethylation of Hcy to methionine (the methionine cycle) predominates over the catabolic degradation of Hcy (trans-sulfuration) because of the order of magnitude difference in Km between MS and CBS [155,156,157,158,159,160]. 

The methylation reactions are necessary for the brain, SAM being the sole methyl group donor in numerous methylation reactions involving proteins, phospholipids, and biogenic amines, and packaging many phospholipids, i.e., polyunsaturated phosphatidylcholines (PC). 

Hyper-homocysteinemia (HHcy) is defined as levels > 15 mol/L, levels between 15 and 30 are considered moderate HHcy, levels at 30–100 micro-mol/L are considered intermediate/severe HHCy, and levels above 100 micro-mol/L are considered as severe (often fatal) HHcy [156,157]. Hcy levels are inversely related to food supplements, principally folate and vitamin B12 [158,159,160], and directly related to smoking, alcohol, physical apathy [161,162], and aging [163]. In vitro studies that explored the correlation between Hcy and inflammation, neurodegeneration, atherosclerosis, and oxidative damage have been inconclusive. Similarly, in vivo trials failed to demonstrate a real benefit in clinical conditions when Hcy is abated by vitamin B12 or B9 supplementation [164].

Genetic causes of severe HHcy linked to a deficiency of CBS or other alterations of remethylation and trans-sulfuration pathways have been reported in neural tube defects and blood–brain barrier alterations [165,166,167,168,169]. Clinical and experimental works demonstrate that HHcy decreases the cell’s methylation potential, modifying the SAM/SAH ratio [170,171,172], and this is the primary determinant for a generalized DNA hypomethylation associated with an excess of oxidative stress [144,170].

Homocysteine accumulation could interfere with endothelium dysregulation, favor oxidative damage, and promote neuroinflammation and neurodegenerative processes [163,171,172,173,174,175]. All these processes occur in SVD; nevertheless, few studies directly focus on HHcy and SVD. Our review attempts to shed some light on the three principal mechanisms of HHcy-induced damage, trying to focus on SVD (Figure 3).

## 4. Homocysteine and Neurodegeneration

HHcy is linked to neurodegeneration, starting from the well-known relationship between its elevation during aging. Many in vivo and in vitro studies showed that HHcy favors the Abeta1–40 deposition in AD [174], mediated by an Hcy-induced upregulation of the Endoplasmic Reticulum Protein (HERP). HERP favors the c-secretase enzyme activity and the consequent increment of the intra- and extra-cellular accumulation of Abeta1–40 and Abeta 42 [175,176,177,178]. 

Hcy is strongly related to neurodegenerative/neuroinflammation conditions by the homocysteinylation process. Homocysteinylation leads to protein damage, i.e., protein denaturation, enzyme inactivation, inflammatory activities, and amyloid-oligomers deposition [179,180,181,182,183,184,185]. Under normal metabolic conditions, the cellular synthesis of Hcy thiolactone is rather low because intracellular concentrations of Hcy are relatively low [186]. If Hcy levels are increased because of a reduction in transmethylation and/or trans-sulfuration, Hcy thiolactone synthesis is enhanced—it could be as much as 60% of the metabolized Hcy [186]. Hcy can be linked to a protein via an isopeptide bond to lysine (Lys) residues (N-Hcy-protein) [187,188,189] or via a disulfide bond to Cys residues (S-Hcy-protein) [190,191,192,193]. N-homocysteinylation is an emerging post-translational protein modification that impairs or alters the protein’s structure/function and causes protein damage [194]. There are two limiting processes of the N-homocysteinylation: the quantity of cyclic Hcy-thiolactone (dependent on HHcy) and the number of lysine residues encountered [195,196,197]. The most evident result of the general homocysteinylation process is protein aggregation and virtual protein misfolding. Thus, Hcy-thiolactone induces apoptosis directly in endothelial cell cultures in in vitro and in vivo models [195]. 

Hcy is also linked to neurodegenerative pathology by influencing tau phosphorylation. As previously described [4,9], tau protein has many functions: the correct assembly of microtubules, directing, therefore, the axonal micronutrients transport toward the neuronal soma. The active form of tau needs constant dephosphorylation mediated by methyltransferase systems (the so-called PPM1 and PPM2A), and the methylation occurs through SAM-dependent reactions [198,199,200,201,202]. Tau hyperphosphorylation has two direct consequences: (1) the disaggregation of microtubules, which leads to an inhibition of axonal transport, and (2) a neuronal death, together with a deposition of damaged microtubules, which forms the so-called tau depositions, or neurofibrillary tangles [203,204,205]. These phenomena have always been associated with degenerative conditions (AD, frontal Pick complex, etc.), but they have also been demonstrated in neuroblastoma cultured cells when the culture medium is depleted by folate, and an increase of P-tau by 66% occurs [206].

HHcy has an intrinsic toxic property [4,9,207] as it acts as an agonist of NMDA (N-methyl D-Aspartate) receptors [208,209,210,211] depending on glycine concentration. Hcy acts as a partial antagonist of the NMDA receptors [4,162,171,207,208], but when the glycine concentration is increased (like in the brain ischemia, in vasospasms, i.e., in prolonged migraine aura attack), even low doses of Hcy could act as an agonist of NMDA channels [212,213], inducing an enhancement of calcium flows [213]. HHCy promotes an extracellular signal-regulated kinase activity in the hippocampus, regulated or blocked by three glutamate receptor antagonists (NMDA, not-NMDA, and metabotropic receptors) [154,214]. It has been suggested that Hcy could directly activate group I metabotropic glutamate receptors, favoring calcium influx currents [212].

Collectively, HHcy exerts essential alteration in the SVD pattern. HHcy induces an increase of Abeta 1–40 toxicity on the smooth muscle cells of the brain’s small arteries, where cerebral amyloid depositions occur, transforming the event into cerebral amyloid angiopathy (CAA), a constant finding in overt SVD condition [4,9,215,216,217]. Moreover, the HHcy condition enhances the m-RNA (Messanger-RNA) production of the C-reactive protein (CRP), over-expressing the NR1 subunit of NMDA receptor expression [4,218]. HHcy enhances the signal pathway cascade, mediated by CRP hyperproduction, mediated by NMDA-ROS-erk1/2/p38-NFK-Beta (NFK = Nuclear Kappa Factor-Beta), which occurs in the smooth muscle cells’ brain small arteries [218]. Homocysteinilation promotes apoptosis [195], endothelium alterations, protein misfolding, and protein aggregation. In fact, the multiple lysine-rich proteins are fibrinogen [196,219], high-density lipoprotein [220], lysine oxidase [221], and cytochrome c [197], and all of them homocysteinylate, aggregate [195], and lead to a general pro-thrombotic condition [196,220,221,222], enhanced coagulation [223], and reduced fibrinolysis [224,225].

## 5. Homocysteine and Neuroinflammation

The pivotal role of HHcy in neuroinflammation is the acceleration of the lipid peroxidation derived from the disruption of the redox system in vascular endothelium, and consequently, among neural cells [226,227,228]. HHcy is always present in multiple traumatic damages, sepsis, multi-organic failure, etc., and HHCy is a sign of poor prognosis [229,230]. 

Animal models showed that HHcy promotes the increase of TNF-alpha, IL1-beta, is inversely associated with a diminution of cystathionine-gamma-lyase-derived H2S in macrophages, and upregulates the transcriptional fibroblast growth factor-2 [9,231,232,233,234]. HHcy directly acts on the endothelium by inducing an upregulation of IL-6, IL-8, TNF-alpha expression [235,236,237], together with cathepsins, involved in the endothelium-inflammatory and vascular remodeling processes [238,239], by influencing IL-6 and TNF-alpha [240,241,242,243,244,245] and enhancing the VEGF/ERK1/2 signaling pathway [240,241,242,243,244,245,246,247], which is a constant in the atherosclerosis process [247]. 

HHCy plays like an agonist of NMDA receptors in CNS (Central Nervous System) and neutrophils and macrophages whenever glycine increases [213]. HHcy activates NMDA receptors, inducing a significant intra-cytoplasmic calcium inflow, with the consequent lipoperoxidation inflammatory process, hyper-activation of the oxidative process accumulation of ROS species [248,249,250]. HHcy also induces a pro-inflammatory status by direct interference with B-control. An in-vitro study demonstrated that there is an upregulation of pyruvate kinase muscle isoenzyme 2 (PKM-2), B-mediated, inhibited by shikonin [251], which mainly promotes the inflammatory basis of atherosclerosis cascade [162,171,251].

HHcy is correlated to a higher quantity of asymmetric dimethylarginine (ADMA), which acts as an inhibitor of eNOS, which catalyzes the production of NO from arginine [252,253,254,255,256]. Together with elevated levels of ADMA, HHcy promotes an increase of the endoplasmic reticulum (ER) stress, upregulating metalloproteinases-9 (MMP-9), and inducing apoptosis [244,255,256,257,258,259,260].

A very new light has been shed on the endothelium effects of HHcy, mediated by ER stress and unfolded protein response (UPR), both events promoting apoptosis in endothelial cells [261,262]. UPR usually upregulates the ER and promotes increased chaperon production, controlling the transcription and translation process, and downregulating the ER proteins [262,263,264,265]. Thus, when there is a hyper-induction of ER stress induced by HHcy [195], there is an accumulation of protein folding capacity. The overwhelming protein accumulation promotes cell modifications, alterations of cell pseudopods, loss of cell adhesion capacity, and caspase-mediated cell death [264,265].

Attention has recently been dedicated to the pro-inflammatory effect of HHcy, exerted directly on smooth muscle cells: HHCy mouse models were found to have enhanced expression of the receptors for the AGEs or vascular cell adhesion molecule [222,223], and MMP-9 [196]. The inflammation cascade could be mediated by the effects on smooth muscle cells rather than on the endothelium alterations [194,264,266,267,268]. The effect of Hcy on B and T cells’ modulation is still undefined, although recent in vitro data suggest B lymphocytes’ activation.

HHcy exerts an overt effect on the global cellular protein quality control (PQC), essential for proteome integrity and cell viability [269]. HHcy has been demonstrated to reduce chaperone levels and impair the UPR systems and control process [269,270,271]. HHcy mice models enhance brain microglia by expressing pro-inflammatory cytokines [272,273,274,275,276,277,278,279,280], particularly the signal transducer and activator of transcription3 (the so-called STAT3). STAT3 helps the microglial regulation of different pro-inflammatory genes [278], such as Il-1-beta, TNF-alpha, and Il-6 [279,280].

## 6. Homocysteine and Oxidative Stress

Oxidative damage is the most accepted consequence of HHcy [9,162,171,220,281,282,283,284,285] and is linked to the oxidation process of the free thiol group of Hcy when it binds many different proteins, such as albumin, other low-weight plasma thiols, or other molecules of Hcy.

Four different mechanisms have been proposed to explain the oxidative stress induced by HHcy [227], not self-excluding: (1) a possible auto-oxidation induced by Hcy, (2) general inhibition of the cellular antioxidant enzymes, (3) NOS-derived production of superoxide anion, through a direct uncoupling of the eNOS, and the disruption of the extracellular superoxide dismutase of the endothelium, and (4) via direct activation of NADPH oxidases [227], that seems to occur directly in the microglia, inducing a hyperactivation of it [286,287]. It has been well-documented that excessive activation of NADPH oxidases contributes to the pathogenesis of numerous peripheral inflammation-related diseases, such as atherosclerosis, diabetes, hypertension, ischemic stroke, and cardiovascular diseases. As a significant superoxide-producing enzyme complex, phagocytic NADPH oxidase (PHOX) is essential for host defense. The discovery of PHOX and non-phagocytic NADPH oxidases in astroglia and neurons further reinforces NADPH oxidases’ critical role in oxidative stress-mediated chronic neurodegeneration [286]. Physiologically, NADPH oxidase-derived ROS have been implicated in the regulation of vascular tone by modulating vasodilation directly (H2O2 may have vasodilator actions) or indirectly by decreasing NO bioavailability (mediated by ·O2− to form ONOO−) [287]. ROS is involved in inflammation, endothelial dysfunction, cell proliferation, migration and activation, fibrosis, angiogenesis, cardiovascular remodeling, and atherosclerosis. These effects are mediated through redox-sensitive regulation of multiple signaling molecules and second messengers, including mitogen-activated protein kinases, protein tyrosine phosphatases, tyrosine kinases, proinflammatory genes, ion channels, and Ca2+ [287].

Trans-sulfuration of homocysteine, catalyzed by the vitamin B6-dependent enzymes, produces cystathionine β-synthase (CBS) and cystathionine γ-lyase (CSE). CBS converts homocysteine and serine into cystathionine, which CSE takes up to generate cysteine. CBS and CSE are also the major enzymes responsible for the biogenesis of hydrogen sulfide (H2S), a gasotransmitter known for its regulatory role in many physiological processes. HHcy causes a decrease in H2S production in mice models, the hippocampus [288], and the cardiovascular system, reducing its cardio-protective effects [288] (Figure 4).

HCy can cause endothelial damage by the effect of lectin-like oxidized low-density lipoprotein receptor-1 (LOX-1) DNA methylation through toll-like receptor 4 (TLR4)/nuclear factor (NF)-κB/DNA methyltransferase (DNMT1) [289], allowing ox-LDL (Oxidized Low-Density Lipoproteins) to accumulate in the sub-endothelial layer and promoting atherosclerotic plaques’ formation [289,290,291,292]. These reactions promote pro-coagulative status (directly mediated by platelet activation and through the N-homocysteinylation of fibrinogen and other pro-coagulative proteins [220,221,222]). Moreover, HHcy (directly and by the HHcy-mediated inhibition of Dimethylarginine dimethylaminohydrolase (DDAH), which causes an ADMA accumulation), induced ROS production decreases NO production and bioavailability, triggering increased redox signaling [293,294,295,296,297].

ROS accumulation’s oxidative stress is the primary mechanism that mediates homocysteine-induced vascular injury in SVD and endothelium dysregulation [298,299,300,301,302]. A study on neuroblastoma cells incubated with HHcy [303] determined different time- and concentration-dependent results [4,162,171,303]. This study suggests the potential genotoxic stress, time-exposure, and Hcy concentration relationship on endothelial and smooth muscle cells [303]. At the very beginning, HHcy induces a correct endothelium response, mediated by the formation of S-nitrose-Hcy, which is an endothelium protector factor [303]. Longer HHcy exposure induced a downregulation of eNOS and provoked oxidative damages [293,304,305,306,307,308,309,310,311,312,313].

## 7. Conclusions

The interplay between HHcy and SVD is relatively novel. Only a few studies have been written during the last six months, defining a potential role of homocysteine inside the complexity of SVD pathogenesis [314,315,316]. A very recent study showed a dose-independent relationship between the plasma Hcy levels and the development of SVD [317]. The study needs to be confirmed in a much larger number of patients. Moreover, a relatively recent study produced contradictory results in coronary stenosis, the prevalence of significant coronary artery stenosis, atherosclerotic, calcified, mixed, and non-calcified plaques increased with homocysteine. However, after adjusting cardiovascular risk factors, there were no statistically significant differences in the adjusted odds ratios for atherosclerotic plaque and mixed plaques between the third and first homocysteine tertiles. In asymptomatic individuals, homocysteine is not associated with an increased risk of subclinical coronary atherosclerosis [318]. These results need better and dedicated new works. While many studies focused on thrombosis and HHcy, HHcy and coronary disease, stroke, and major vessel disease, few data are available on HHcy and vascular and neurodegeneration because SVD in the brain is a relatively recent entity. SVD is a complex clinical entity linked to the aging modification of the small arteries, altered endothelium activation, oxidative damage, and generally by a chronic inflammatory state induced by persistent hypoperfusion. Inflammation, oxidative damages, misfolding, and neurodegeneration happen together in a dynamic sequence during the development of SVD. Hcy’s role could change in the temporal sequences of events. Definition of the different roles of Hcy at the different cellular levels, promotion of the confluency of altered white matter areas, and times of the development of SVD in the brain may provide hints as to the modulation of Hcy to prevent disease.

## Figures and Tables

**Figure 1 ijms-22-02051-f001:**
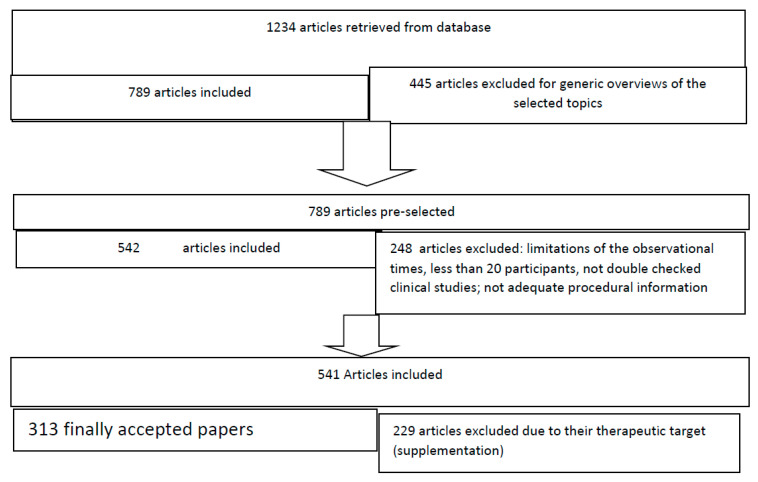
Flowchart of search strategy and selection criteria.

**Figure 2 ijms-22-02051-f002:**
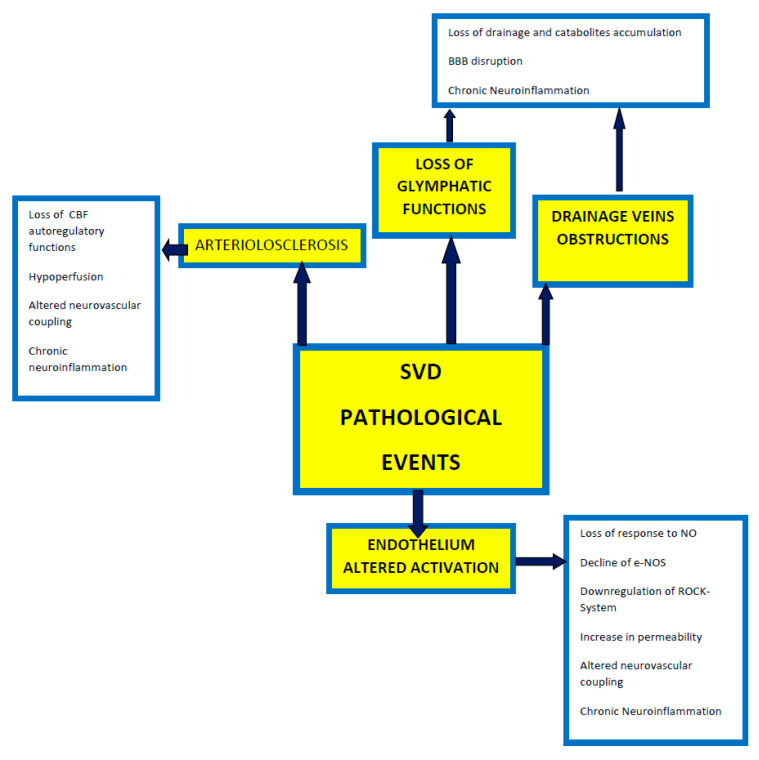
A synoptical overview of the pathological events which occur in SVD (Small Vessel Disease) (abbreviations: BBB = blood–brain barrier; eNOS = endothelium-derived NO synthase; NO = Nitric oxide; ROCK = Rho-associated protein kinase; SVD = Small Vessel Disease).

**Figure 3 ijms-22-02051-f003:**
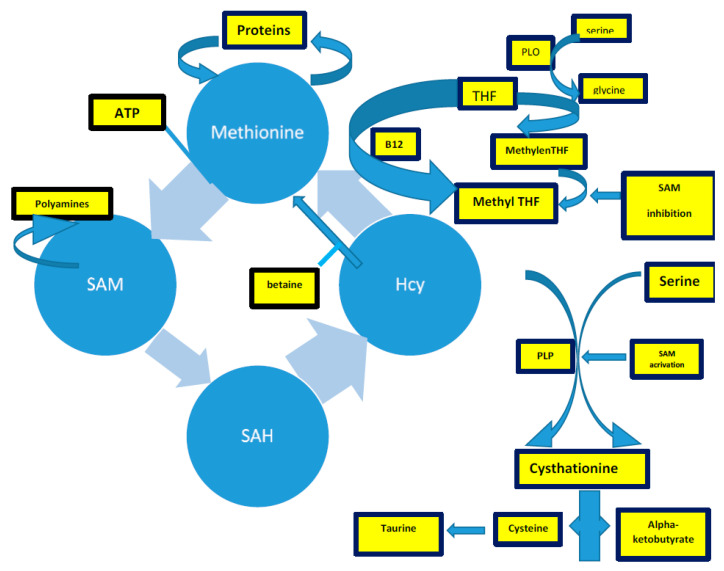
The complex of Hcy production, described in the text. Acronyms: SAM: s-adenosylmethionine; THF: tetrahydrofolate; PLP: pyridoxal-5-phosphate.

**Figure 4 ijms-22-02051-f004:**
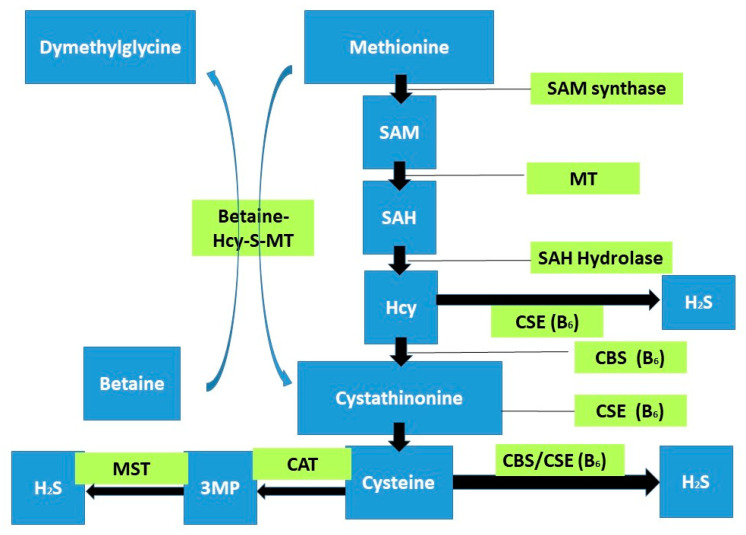
A schematic overview of the association between homocysteine and H2S is represented. Homocysteine is biosynthesized from methionine by S-adenosyl methionine (SAM) synthetase, methyltransferase (MT), and s-adenosyl-L-homocysteine (SAH) hydrolase. Hcy can be either remethylated to methionine (see Figure 2) or trans-sulfurated to cysteine under the catalysis of cystathionine beta-synthase (CBS) and cystathionine gamma-lyase (CSE) that requires vitamin B6 as a cofactor. Hcy and cysteine are substrates for H2S production, and the generation of H2S is catalyzed by CBS, CSE, and 3-mercaptopyruvate sulfurtransferase (MST).

## Data Availability

The data presented in this study are openly available in all the articles cited in references. Data sharing not applicable.

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
