# Peer review of "Homocysteine in Neurology: A Possible Contributing Factor to Small Vessel Disease"

_ijms, 2021, doi:10.3390/ijms22042051_

Round 1

Reviewer 1 Report

Many articles about homocysteine accumulation in vascular or degenerative neurological diseases were published, but the results are still undefined. Authors in this review summarize the role of homocysteine in the development of small vessel disease (SVD) and neurological damage. This manuscript reviews more than 300 articles and provide a complex survey of current literature dealing with this topic. The topic of this manuscript is up to date, interesting and well suited for International Journal of Molecular Sciences. The manuscript is well written and divided into 7 parts, the text is clear and easy to read. For better understanding authors used 4 illustrations and also added acronyms list to the end of the manuscript. This aid to the readers understanding. I suggest checking for some small spelling mistakes and grammar errors. Otherwise, I have no major concerns about this manuscript and I recommend it for publication.

I recommend the manuscript for publication.

Author Response

Dear Sir,

Thank you for your review and for your words.

the manuscript has been re-read, some parts have been rephrased in a clearer set, and typos have been checked and corrected.

Thank you again

Reviewer 2 Report

The authors reviewed a possible relationship between homocysteine levels and small vessel disease

The review is almost well written and very interested.

Please correct many typographical error such as space, double line etc…

Please correct a the graphical metabolism ( a box is empty) and add all the abbrevations.

The authors should be better discuss the data before to declare that “The interplay between HHcy and

SVD is relatively novel. A very recent study showed a dose-independent relationship between the

plasma Hcy levels and the development of SVD” because Park et al declare for example that Homocysteine is not a risk factor for subclinical coronary atherosclerosis in asymptomatic individuals (DOI: 10.1371/journal.pone.0231428)

The authors should expand the reference sections maybe incorporating other articles to increase the strength of the paper (as for example DOI 10.3390/metabo11010037 or 10.1016/j.ijcha.2020.100515 or 10.1161/JAHA.120.017746.

Author Response

Dear Sir, 

thank you for your review. The ms has been revised, and typos and other mistakes have been checked and corrected. 

We have completed the discussion with your suggestion; we focused on Park et al, and on the strong need for more studies, considering the interlocutory results obtained by that group. We have cited :Cordaro M, Siracusa R, Fusco R, Cuzzocrea S, Di Paola R, Impellizzeri D. Involvements of Hyperhomocysteinemia in Neurological Disorders.  2021 Jan;11(1) . doi:10.3390/metabo11010037.; Toya T, Sara JD, Lerman B, Ahmad A, Taher R, Godo S, Corban MT, Lerman LO, Lerman A. Elevated plasma homocysteine levels are associated with impaired peripheral microvascular vasomotor response. Int J Cardiol Heart Vasc. 2020 Apr 17;28:100515. doi: 10.1016/j.ijcha.2020.100515. ;  Ahmad A, Corban MT, Toya T, Sara JD, Lerman B, Park JY, Lerman LO, Lerman A. Coronary Microvascular Endothelial Dysfunction in Patients With Angina and Nonobstructive Coronary Artery Disease Is Associated With Elevated Serum Homocysteine Levels. J Am Heart Assoc. 2020 Oct 20;9(19):e017746. doi: 10.1161/JAHA.120.017746.

We have tried to expand the discussion on these points.

We have rewritten figure 3; we have added all the abbreviations.